# Molecular Characterization and Therapeutic Opportunities in KRAS Wildtype Pancreatic Ductal Adenocarcinoma

**DOI:** 10.3390/cancers16101861

**Published:** 2024-05-13

**Authors:** Aakash Desai, Alexander H. Xiao, Daheui Choi, Merih D. Toruner, Daniel Walden, Thorvardur R. Halfdanarson, Steven Alberts, Robert R. McWilliams, Amit Mahipal, Daniel Ahn, Hani Babiker, Gulnaz Stybayeva, Alexander Revzin, Sani Kizilbash, Alex Adjei, Tanios Bekaii-Saab, Aaron S. Mansfield, Ryan M. Carr, Wen Wee Ma

**Affiliations:** 1Department of Oncology, Mayo Clinic, Rochester, MN 55902, USA; adesaimd@uab.edu (A.D.); merih_toruner@brown.edu (M.D.T.); halfdanarson.thor@mayo.edu (T.R.H.); alberts.steven@mayo.edu (S.A.); mcwilliams.robert@mayo.edu (R.R.M.); kizilbash.sani@mayo.edu (S.K.);; 2Department of Medicine, Mayo Clinic, Rochester, MN 55902, USA; xiao.alexander@mayo.edu; 3Department of Physiology and Biomedical Engineering, Mayo Clinic, Rochester, MN 55902, USA; choi.daheui@mayo.edu (D.C.); stybayeva.gulnaz@mayo.edu (G.S.); revzin.alexander@mayo.edu (A.R.); 4Division of Hematology and Oncology, Mayo Clinic, Phoenix, AZ 85054, USA; walden.daniel@mayo.edu (D.W.); ahn.daniel@mayo.edu (D.A.); bekaii-saab.tanios@mayo.edu (T.B.-S.); 5Department of Oncology, University Hospitals Seidman Cancer Center, Cleveland, OH 44106, USA; amit.mahipal@uhhospitals.org; 6Division of Hematology and Oncology, Mayo Clinic, Jacksonville, FL 32224, USA; babiker.hani@mayo.edu; 7Taussig Cancer Institute, Cleveland Clinic, Cleveland, OH 44195, USA; adjeia2@ccf.org

**Keywords:** pancreatic ductal adenocarcinoma, KRAS wildtype, MET fusion, vebreltinib, APL-101, PLB1001

## Abstract

**Simple Summary:**

Pancreatic cancer is predicted to be the second-highest cause of cancer mortality in the US by 2040. It is driven by various mutations including *KRAS*, *TP53*, *SMAD4*, and *CDKN2A.* Roughly 1 in 10 patients with pancreatic cancer have wildtype *KRAS (KRAS*^WT^). We studied 27 patients with *KRAS*^WT^ to better understand their molecular characteristics and potential for precision medicine. Our findings revealed that *KRAS*^WT^ PDAC is enriched with potentially treatable genetic alterations, including those affecting the MAPK pathway, DNA repair pathway, and kinase fusion genes. One of our *KRAS*^WT^ PDAC patients was found to have a specific, targetable TFG-MET mutation and they responded well to treatment with a cMET inhibitor. This suggests that *KRAS*^WT^ PDAC has unique genetic characteristics that could be targeted with specific treatments tailored to each patient, highlighting the importance of comprehensive genetic profiling in *KRAS*^WT^ PDAC.

**Abstract:**

Purpose: To investigate the molecular characteristics of and potential for precision medicine in KRAS wildtype pancreatic ductal adenocarcinoma (PDAC). Patients and Methods: We investigated 27 patients with *KRAS*^WT^ PDAC at our institution. Clinical data were obtained via chart review. Tumor specimens for each subject were interrogated for somatic single nucleotide variants, insertion and deletions, and copy number variants by DNA sequencing. Gene fusions were detected from RNA-seq. A patient-derived organoid (PDO) was developed from a patient with a *MET* translocation and expanded ex vivo to predict therapeutic sensitivity prior to enrollment in a phase 2 clinical trial. Results: Transcriptomic analysis showed our cohort may be stratified by the relative gene expression of the KRAS signaling cascade. The PDO derived from our patient harboring a *TFG-MET* rearrangement was found to have in vitro sensitivity to the multi-tyrosine kinase inhibitor crizotinib. The patient was enrolled in the phase 2 SPARTA clinical trial and received monotherapy with vebrelitinib, a c-MET inhibitor, and achieved a partial and durable response. Conclusions: *KRAS*^WT^ PDAC is molecularly distinct from *KRAS*^MUT^ and enriched with potentially actionable genetic variants. In our study, transcriptomic profiling revealed that the KRAS signaling cascade may play a key role in *KRAS*^WT^ PDAC. Our report of a *KRAS*^WT^ PDAC patient with TFG-MET rearrangement who responded to a cMET inhibitor further supports the pursuit of precision oncology in this sub-population. Identification of targetable mutations, perhaps through approaches like RNA-seq, can help enable precision-driven approaches to select optimal treatment based on tumor characteristics.

## 1. Introduction

Pancreatic ductal adenocarcinoma (PDAC) is predicted to be the second-highest cause of cancer mortality in the United States by 2040 [1]. Despite scientific advances, the five-year overall survival has improved only marginally over the last few decades [2]. The majority of patients have advanced disease at presentation that is invariably lethal [3]. Comprehensive molecular and genomic profiling continues to identify potentially ‘druggable’ molecular alterations in PDAC. Highly recurrent pathogenic mutations in PDAC include *KRAS*, *TP53*, *SMAD4*, and *CDKN2A* [4]. In particular, *KRAS* plays a significant role in pancreatic cancer due to the dysregulation of key cellular processes including glycolysis, autophagy and macropinocytosis by altering cellular metabolism and supporting the tumor microenvironment [5,6,7]. Common molecular drivers found in other malignancies are also identified in PDAC but are relatively infrequent [8]. Microsatellite instability is rare in PDAC (<1%), portending the limited role of immune checkpoint inhibitors [9]. Similarly, germline variants in *BRCA1/2* genes, which predict susceptibility to PARP inhibitors, are found in ≤10% of PDAC cases [10], and the response rate to PARP inhibitors is only 20% [11]. Other actionable mutations are even rarer, including those in *NTRK1-3* (<1%), *BRAF* (1–2%), or *ALK* rearrangements (0.16%) [12]. Interestingly, recent PDAC genomic analyses suggest that *KRAS* wildtype (*KRAS*^WT^) PDAC has a higher likelihood of harboring molecularly actionable alterations (8 to 10%) [13]. Drivers in these cases tend to involve MAPK activation (~30–40%), kinase fusions (~40%), or microsatellite instability (~10–20%). Theoretically, these drivers could be selectively targeted and/or amenable to immunotherapy. Compared to *KRAS* mutated (*KRAS*^MUT^) PDAC, the survival for *KRAS*^WT^ disease appears to be superior [14]. Taken together, *KRAS*^WT^ PDACs seem to be a molecularly, if not biologically, distinct subgroup, and comprehensive characterization is needed to discover clinically actionable alterations. We report the evaluation of a cohort of *KRAS*^WT^ PDAC patients at our institution, including the clinical outcome of one patient with *TFG-MET* rearrangement treated using the selective MET inhibitor vebrelitinib (also known as APL-101, PLB-1001, ClinicalTrials.gov #NCT03175224).

## 2. Materials and Methods

### 2.1. Study Design

The study was approved by the Institutional Review Board of Mayo Clinic. We identified 241 subjects with PDAC who underwent CLIA-certified Next-Generation Sequencing (NGS) testing at Mayo Clinic between 1 December 2018 and 1 December 2021. Of these, 27 subjects with *KRAS*^WT^ PDAC were identified and included in the study cohort. Two physicians (A.D. and A.X.) reviewed the medical records and extracted information on clinical data and disease characteristics.

### 2.2. Sample Processing and NGS with Mutational Analysis

NGS was performed by third-party companies, Tempus and CARIS. Prior to testing, an expert pathologist assessment was performed to assess cellularity as a ratio of tumor to normal nuclei. Chemagic 360 sample-specific extraction kits were used for nucleic acid extraction from formalin-fixed paraffin-embedded tissue sections. RNA was purified from the total nucleic acid by DNase-I digestion. xT panel DNA and RNA library construction and sequencing DNA sequencing of 648 genes and whole-transcriptome RNA sequencing were performed as previously described [15].

### 2.3. RNA Sequencing Analysis

RNA-seq FASTQ files were checked for quality using FASTQC (Version 0.11.9) and the paired-end files were aligned to the HG38 reference genome using STAR (Version 2.7.8a). The number of reads was obtained using HTSeq-count (Version 0.9.1). The samples were normalized with the median of ratios using the DeSeq2 package (Version 2.11.40.7) and clustered with Complete-linkage clustering according to the top 1000 most variable genes in the dataset, using Pheatmap R package (Version 1.0.12). Gene Set Enrichment Analysis was performed using the normalized counts and GSEA (Version 4.1.0) with the Hallmarks gene set [16]. Pathways that had less than 0.05 nominal *p*-value were considered significantly enriched gene sets.

### 2.4. Fusion Diagram

The graph of the TFG-MET fusion was made with the Fusion Editor function in ProteinPaint. The fusion coordinates detected by sequencing were submitted according to their codon coordinates to create the graph.

### 2.5. Subject Specimens

A patient-derived organoid (PDO) was developed from a subject with PDAC harboring a *TFG-MET* translocation for ex vivo drug studies. This study was approved by the Institutional Review Board under protocol 17-003174.

### 2.6. Patient-Derived Organoid (PDO) Generation from Biopsies

Generation of PDOs, culture, and drug studies were performed as previously described [17]. Briefly, ultrasound-guided PDAC biopsies were collected and placed onto a 60 mm Petri dish on ice in KRB buffer (Sigma-Aldrich, K4002, St. Louis, MO, USA) and minced into small pieces. Then, the chopped tissue was treated with 2.5 mg/mL collagenase type IV (Sigma-Aldrich, C4-BIOC) and penicillin/streptomycin (Gibco, 15140122, Grand Island, NY, USA) in KRB solution. The mixture was placed into a water bath at 37 °C for 5 min under agitation to digest the extracellular matrix. The extracted cells were collected by filtration in a 100 µm strainer and centrifugated at 1500 rpm for 5 min. To remove red blood cells in the cell pellet, 3 mL of RBC lysis buffer (Invitrogen, 00-4333-57, Carlsbad, CA, USA) was added for 3 min and deactivated with FBS (Gibco, 16000044). Finally, the extracted PDAC cells were seeded onto Matrigel-coated (Corning, 356255, Corning, NY, USA) 12-well plates in organoid media and cultured for 1 week until organoid formation. For subculture, the Matrigel and organoids were broken down using a gentle cell dissociation reagent, and the organoids were freshly embedded with 50% Matrigel-50% organoid media in a well plate. For chemotherapy, organoids were used within four passages.

### 2.7. PDO Materials

Advanced DMEM (12491023), 4-(2-hydroxyethyl)-1-piperazineethanesulfonic acid (15630080), Glutamax (35050061), and penicillin/streptomycin were purchased from Gibco. N2 supplement (AR009), B27 supplement (AR008), epidermal growth factor (236-EG), fibroblast growth factor 10 (345-FG) and Cultrex PathClear Reduced Growth Factor BME Type 2 (3533-010-02) were from R&D systems. N-acetyl-L-cysteine (A7250), Nicotinamide (N0636), Gastrin I human (G9020), NaCl (S5886), D-glucose (G8270), KCl (P5405), and Collagenase type IV (C4-BIOC) were purchased from Sigma-Aldrich. Gentle cell dissociation reagent (07174) was purchased from STEMCELL Technologies (Vancouver, BC, Canada). Gemcitabine and crizotinib (HY-50878) were purchased from pharmacies at the Mayo Clinic and MedChemExpress, respectively. The composition of organoid media is as described by prior studies [18].

### 2.8. PDO Drug Studies and Viability Assessment

PDAC organoids were embedded in 50% Matrigel-50% organoid media and seeded on a 48-well plate. The cells were cultured for 7 days until the organoids generated and proliferated. For drug treatment, the gemcitabine and crizotinib (0.01–10 µM) in organoid media were treated to each well and cultured for 7 days. The negative control is organoid media-treated cells. At the end point (day 7), the media and drugs were removed, and organoids were stained using a live–dead staining kit, which can stain live cells with green fluorescence and dead cells with red fluorescence, respectively. The live and dead cell area is measured by ImageJ analysis. The viability of each organoid was obtained based on the following equation (*n* = 3). Viability % = 100 × A_green_/(A_green_ + A_red_).

## 3. Results

### 3.1. Subject Population

In total, 241 subjects with PDAC at Mayo Clinic between 1 December 2018 and 1 December 2021 were considered for analysis. Of these, 27 (11%) were found to have *KRAS*^WT^ PDAC. The median age at diagnosis was 61 years (range 24–84 years) and 68% were males. In total, 8 (42%) had Stage IV disease at diagnosis with 15 (79%) additional patients ultimately developing metastatic disease (Table 1).

The median length of follow-up from diagnosis was 37 months. Among patients treated for metastatic disease (*n* = 13), the median number of lines of treatment was 3 (range: 0–4). In this cohort, five patients received FOLFIRINOX with a median overall survival (mOS) of 17 months, five received gemcitabine and nab-paclitaxel with mOS of 15 months, and one received pembrolizumab without progression thus far (Table 1). Two patients did not receive any treatment and pursued hospice.

### 3.2. Genomic Characterization

NGS data were collected for genomic characterization of the *KRAS*^WT^ cohort. Among the subjects, four (15%) did not have alterations identifiable by NGS and recognized as biologically relevant. Prior work has found the most common recurrent drivers of PDAC included mutations in *TP53*, *CDKN2A*, and *SMAD4* with frequencies of 64%, 21%, and 17%, respectively [4]. In this *KRAS*^WT^ cohort, mutations in these three genes were seen at frequencies of 41%, 7%, and 4%, respectively (Figure 1).

Chromosomal rearrangements were identified in five (19%) subjects with four (15%) being potentially therapeutically actionable including *CADPS2*-*BRAF* [18], *GP2*-*ERBB2* [19], *EML4*-*NTRK3* [20] and *TFG*-*MET* [21].

Of the 21 (78%) KRAS^WT^ patients for whom microsatellite instability (MSI) status was available, 2 (7%) were MSI-high, and both had an elevated tumor mutation burden (TMB) of 28.4 and 23.7 mutations per megabase (m/MB), suggesting potential sensitivity to immune checkpoint inhibitor pembrolizumab [9]. All other patients had TMB < 10 m/MB. Both MSI-high subjects had pathogenic variants in mismatch repair genes *MSH3*, with one subject with a variant in *MSH2* and another subject in *MSH6*.

Four subjects (15%) harbored pathogenic variants in genes important in DNA damage response. Two harbored mutations in *BRCA2* while one had a mutation in *PALB2*, which have been associated with increased sensitivity to platinum-based chemotherapy and potentially poly-ADP ribose polymerase (PARP) inhibitors [11].

Other identified mutations are predicted to result in altered WNT signaling (15%), PI3K/AKT signaling (15%), and MAPK/RAS signaling (4%). Epigenetic regulators were particularly enriched in the two individuals with MSI-high disease. The most commonly mutated chromatin remodelers were SWI/SNF components (*ARID1A* and *ARID1B*, 7.4%) as well as the histone 3 lysine 4 methyltransferase *KMT2D* (7.4%). These are also amongst the most recurrently mutated genes in *KRAS*^MUT^ PDAC [4].

### 3.3. Transcriptomic Characterization

Global RNA-sequencing data were available for 19 (70.4%) of the subjects in the *KRAS*^WT^ cohort. RNA-seq was performed on microdissected tumor tissue. Unsupervised non-hierarchical clustering of transcriptomic profiles resulted in two distinct groups of patients (Figure 2A). Gene set enrichment analysis (GSEA) was used to further characterize differences in gene expression between these groups. This revealed the enrichment of HALLMARK_KRAS_SIGNALING_DN in Group 2 while HALLMARK_KRAS_SIGNALING_UP was enriched in Group 1 (Figure 2B). This gene set includes 200 genes up-regulated by KRAS activation [22]. A heatmap comparing the relative expression of some of these most differentially expressed genes in the KRAS_SIGNALING_UP gene set is depicted in Figure 2C. There does not appear to be a clear mutational profile logically predicting these transcriptomic subtypes.

### 3.4. Translational Studies

To investigate the potential for precision medicine in the *KRAS*^WT^ PDAC cohort, a patient-derived organoid (PDO) was generated from a biopsy sample of a male subject’s disease. Figure 3 illustrates correlative studies of this subject with a *MET* translocation. *MET*, which encodes for the hepatocyte growth factor receptor protein, plays a vital role in embryogenesis and tissue homeostasis. *MET* dysfunction is important in the pathogenesis of many cancers. Within pancreatic cancer, *MET* dysregulation is associated with more aggressive disease [23]. The translocation involved *TFG* at exon 6 (3:100455560) and *MET* at exon 15 (7:116414935) confirmed by both Tempus and SPARTA Caris confirmatory testing (Figure 3A). While the translocation truncates the *MET* gene, leading to the loss of exon 14 as depicted, the catalytic domain remains intact in the predicted gene product. Given this mutation may be associated with increased *MET* signaling, dose–response experiments were performed to predict differential therapeutic sensitivity. While the PDO was resistant to gemcitabine across doses, it had significant sensitivity to multi-tyrosine kinase inhibitor crizotinib (Figure 3B,C). Based on the results of the above experiment, the patient was enrolled in the phase 2 SPARTA clinical trial (NCT03175224) which involved assessing the efficacy of vebreltinib (also known as APL-101, PLB-1001, and bozitinib), an oral inhibitor of *MET*. Treatment involved vebreltinib at 200 mg by mouth twice daily. Of note, this patient had previously progressed on FOLFIRINOX, gemcitabine with nab-paclitaxel, and a nanoliposomal irinotecan regimen. Images depicted in Figure 3D are coronal sections from computed tomography (CT) studies performed before (above) and after (below) initiation of vebreltinib. The baseline scan revealed a complex necrotic mass centered in the pancreatic head and measured 9.3 × 8.3 cm. The mass abuted the distal stomach and proximal duodenum. Tumor shrinkage was observed after approximately two months of therapy (with the pancreatic mass decreasing to 4.0 × 3.9 cm). The patient achieved partial response by RECIST criteria after approximately four months of treatment. The patient remains on vebreltinib without evidence of disease progression as of July 2023 (18 months post enrollment).

## 4. Discussion

We describe a clinical cohort of subjects with *KRAS*^WT^ PDAC (*n* = 27), demonstrating the molecular underpinnings and potential role for precision therapeutics in this select population. Of the 241 subjects at Mayo Clinic with PDAC considered for this study, approximately 11% were found to be *KRAS*^WT^, which is consistent with previous estimates [14].

Tumor specimens for each subject were interrogated for somatic single nucleotide variants, insertion and deletions, and copy number variants by DNA sequencing. Gene fusions were detected from RNA-seq in an unbiased and comprehensive manner. Our cohort demonstrates the enrichment of KRAS^WT^ PDAC for therapeutically targetable molecular alterations as suggested by previous reports [24]. In the context of current existing treatments, we identified targets across the MAPK pathway (ERBB2, BRAF), DNA repair pathway (PALB2, ARID1A/B, MSH2/6), and kinase fusion genes (MET, NTRK). The most altered gene in *KRAS*^WT^ PDAC was *TP53* (Figure 1) which occurred more frequently compared to previous reports in *KRAS*^MUT^ PDAC. Alterations in *TP53* currently lack approved therapies, though many are under development currently [25].

*BRAF*, a downstream effector of *KRAS,* was altered in 3/27 (11%) of subjects, with two *BRAF*-based fusions present. These fusions are known to dimerize and result in elevated kinase activity, highlighting the importance of the MAPK kinase pathway in PDAC biology. Targeting BRAF fusions clinically with an MEK inhibitor or BRAF inhibitors has been reported in multiple tumor types [18]. *BRAF* alterations have been found to occur in similar frequencies in *KRAS*^WT^ PDAC in other reports, which have also described this to be mutually exclusive with *KRAS*^MUT^ PDAC [14].

Importantly, we also demonstrate proof-of-concept of targeting molecular underpinnings in a subject with KRAS^WT^ PDAC driven by *TFG-MET* fusion where PDO showed sensitivity to crizotinib in vitro. This was further confirmed by a durable, partial response following enrollment of the subject in a basket trial of the oral MET inhibitor vebreltinib. This case emphasizes the importance of comprehensive multigene tumor profiling in PDAC and the potential benefit to subjects with actionable mutations. It is, to our knowledge, the first case of *TFG-MET* fusion displaying oncogenic driver activity, then clinically validated by partial response to a selective MET inhibitor.

The presence of KRAS signaling enrichment in a significant proportion of our KRAS WT cohort suggests there are a number of alternative genotypes conferring a similarly KRAS-driven phenotype in PDAC.

While mutation-specific KRAS inhibitors likely have no role in the management of KRAS WT PDAC, RMC-6236, an RAS^MULTI^ inhibitor that targets the active form of RAS proteins may be useful in some of the presented cases. For example, in the presented TFG-MET translocated (Case 3) case, MET downstream signaling activates signaling molecules including RAS. Thus, when oncogenic signaling in PDAC WT cases is upstream of RAS signaling, concomitant use of multikinase KRAS (on) inhibitors may be useful. In our cohort, this may also apply to the GP2-ERBB2 translocation (Case 2), ERBB2 missense mutation (Cases 10 and 11), and EML4-NTRK3 translocation (Case 4), cases where downstream KRAS serves as a common effector of oncogenic signaling. Case 14 demonstrates loss of APC, which confers increased WNT signaling. Interestingly, this has been shown to result in de-repression of KRAS signaling [26]. Thus, this case may also benefit from multikinase KRAS (on) inhibitors, which could have significant implications for the management of colorectal cancer where pathogenic APC mutations are enriched. However, Case 6 is an example where such inhibitors may not be effective. While the GSEA similarly identified increased KRAS signaling in this case, this tumor harbors a BRAF mutation, downstream of KRAS.

Routine molecular profiling of PDAC is established to identify *BRCA* mutations and MMR deficiency [27]. However, certain gene fusions have been described as effective therapeutic targets in PDAC, such as our subject’s TGF-MET fusion [12]. With various MAPK pathway targeting strategies currently in development, there is an opportunity to harness the numerous alterations seen in the KRAS^WT^ PDAC associated with significant MAPK activation [28].

Furthermore, our subjects with KRAS^WT^ PDAC have microsatellite instability in ~8% of cases, comparable to previous studies [14]. This may suggest that immunotherapy may have potential activity in the KRAS^WT^ PDAC cohort (given the higher frequency of MSI-H/dMMR and TMB-high tumors) compared to non-biomarker-selected PDAC [29]. Interestingly, only two cases (8%) harbored *SMAD4* loss, typically found mutated in ~20% of *KRAS*^MUT^ PDAC.

Our study confirms the increasing evidence suggesting that there are systematic differences between KRAS^WT^ PDAC and *KRAS*^MUT^ PDAC tumors at the genomic level that result in altered transcriptomic profiles. This likely drives the distinct differences in clinical behavior between these two disease subtypes. In agreement with this, our real-world evidence shows prolonged survival of subjects with *KRAS*^WT^ tumors, consistent with previously reported smaller series [30]. Molecular features that potentially contribute to improved survival include higher frequency of MSI-H/dMMR and associated increased lymphocyte infiltration [31]. In addition, it is conceivable that the slightly prolonged overall survival observed in the platinum-treated subjects compared to the gemcitabine–nab-pacltiaxel-treated subjects may be a result of a higher rate of DNA damage response genes mutations, including *PALB2* and *BRCA2*, in the *KRAS*^WT^ cohort.

## 5. Conclusions

In conclusion, patients with *KRAS*^WT^ PDAC represent a distinct subgroup who may benefit from comprehensive molecular profiling to identify targetable oncogenic drivers for clinical trial participation or treatment with targeted therapies with tumor agnostic approvals. Identification of targetable mutations, perhaps through approaches like RNA-seq, can help enable precision-driven approaches to select optimal treatment based on tumor characteristics, such as the presence of a MET alteration. Collectively, multigene profiling, including the determination of the *KRAS* status as part of the initial diagnostic workup, should be considered in the routine management of PDAC, and not just for *BRCA* or MMR deficiency testing. Future studies should focus on the identification of predictive transcriptomic signatures, the role of liquid biopsy in personalizing treatment and biomarker-selected treatment strategies in *KRAS*^WT^ PDAC.

## Figures and Tables

**Figure 1 cancers-16-01861-f001:**
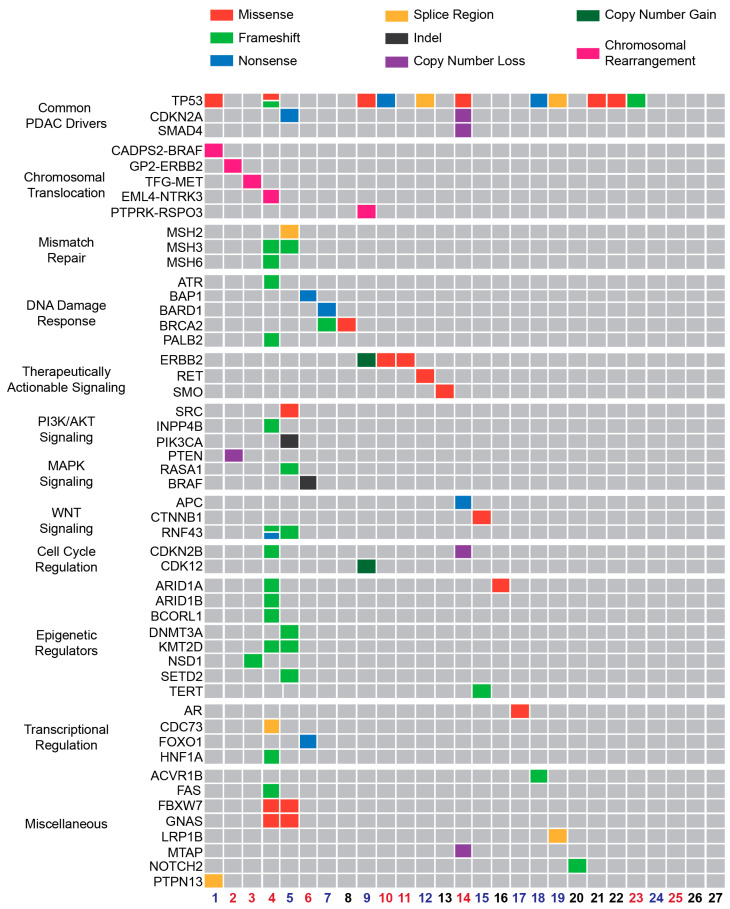
Tumor genotyping of *KRAS*^WT^ PDAC reveals potentially therapeutically actionable mutations. Depicted is an Oncoprint demonstrating the presence and type of mutation detected on NGS of the *KRAS*^WT^ PDAC cohort. Each column represents an individual subject. General groups of genes are separated and labeled on the left. The colored subject numbers are indicative of the patient’s transcriptomic subgroup with red representing Group 1 and blue representing Group 2 (see Figure 2). Black means that RNA-sequencing data was not available or not performed.

**Figure 2 cancers-16-01861-f002:**
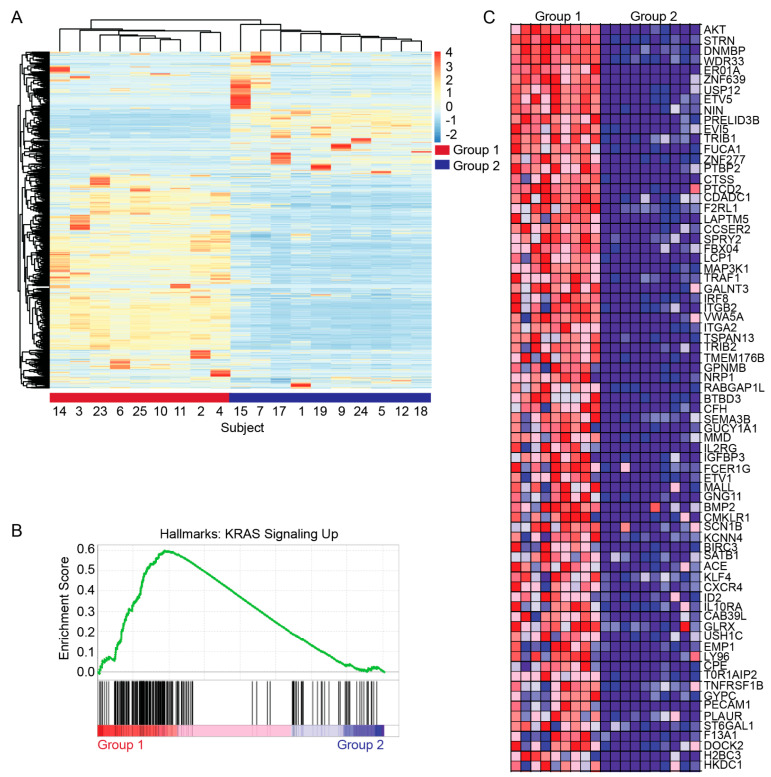
Transcriptomic characterization of *KRAS*^WT^ PDAC reveals clustered subtypes distinguished by differential expression of genes regulated by KRAS activity. (**A**) RNA-sequencing data represented as a heat map of the top 1000 most differentially expressed genes. The dendrogram (top) represents unsupervised non-hierarchical clustering of the subjects’ gene expression. (**B**) Gene set enrichment analysis comparing Group 1 and Group 2 showing differential enrichment of the HALLMARK_KRAS_SIGNALING_UP gene set. (**C**) Heatmap comparing individual relative expression of genes included in the KRAS signaling gene set. Each column represents data from an individual subject.

**Figure 3 cancers-16-01861-f003:**
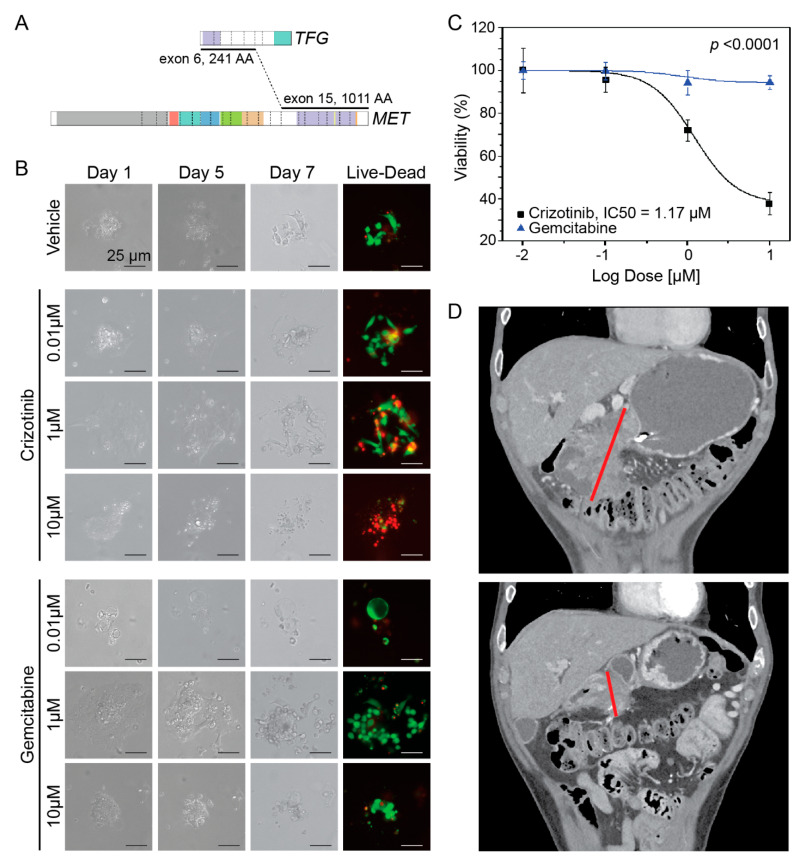
A PDO harboring a *TFG*-*MET* translocation serves as a successful patient avatar predicting sensitivity to muti-tyrosine kinase inhibitors. (**A**) Diagram illustrating the translocation breakpoints between the *TFG* and *MET* loci. Colored areas are representative of predicted functional domains. The purple C-terminal domain at the *MET* locus encodes the protein’s catalytic domain. The translocation involves *TFG* at exon 6 (3:100455560) and *MET* at exon 15 (7:116414935). AA = amino acid. (**B**) Representative images from dose–response experiments of a PDO harboring the *TFG-MET* translocation. Phase contrast images (40× magnification) were taken on days 1, 5 and 7 of treatment with the indicated increasing doses of crizotinib (above) and gemcitabine (below). Immunofluorescence images taken on Day 7 were after live–dead staining in which live cells have green fluorescence while dead cells fluoresce red. (**C**) Dose–response curve comparing percentage cell viability between crizotinib and gemcitabine (*n* = 3). Data are normalized to vehicle control. (**D**) Coronal computed tomography images of the *KRAS*^WT^ pancreatic head mass harboring the *TFG-MET* translocation. The red line indicates the largest diameter of the tumor. Images depict the tumor at baseline (above) and two months after initiation of vebreltinib (below).

**Table 1 cancers-16-01861-t001:** Patient characteristics and clinical outcomes.

Median Age (Years)				61 (Range 24 to 84)				
Male Sex (%)				68				
Stage IV at Diagnosis (%)				42				
Development of Metastatic Disease				15				
Median Follow-up (months)				23 (Range 0.6 to 142.9)				
MSI Status								
High (%)				2/21 (9.5)				
Low (%)				19/21 (90.5)				
TMB Status								
High (>10 mutations/MB) (%)				2/21 (10.5)				
Low (>10 mutations/MB) (%)				19/21 (89.5)				
Treatment	Patient	Sample Number	Baseline Measurement	Best Response	% Change	Best Response	Objective Response Rate	Median Overall Survival in Months
FOLFIRINOX	Subject 1	1	2.5 cm	3.3 cm	+32%	PD	25%	17
	Subject 2	8	2 cm	1 cm	−50%	PR		
	Subject 3	10	11.1 cm	3.2 cm	−72%	PR		
	Subject 4	17	3.7 cm	2.9 cm	−21.6%	SD		
Gemcitabine and nab-paclitaxel	Subject 1	-	5.4 cm	5.2 cm	−3.7%	SD	0%	15
	Subject 2	12	2.8 cm	5 cm	+78%	PD		

## Data Availability

The data generated in this study are available upon request from the corresponding author.

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
