# Peer review of "Molecular Characterization and Therapeutic Opportunities in KRAS Wildtype Pancreatic Ductal Adenocarcinoma"

_cancers, 2024, doi:10.3390/cancers16101861_

Round 1

Reviewer 1 Report

Comments and Suggestions for Authors

A timely research article by Dr. Ma and the group elaborates on WT KRAS's role in PDAC with molecular characterization and therapeutic opportunities. This article has an immense impact on the translational aspect. However, few things must be addressed before it is ready for acceptance. They are as follows: 

1. KRAS  mutation, a hallmark of PDAC, is linked to critical aspects of its biology, such as inflammation, immune evasion, and altered metabolism (PMID: 33870211, PMID: 38471457). Authors must add a line in the introduction by mentioning this with referred references.

2. In Figures 1 and 2, the words next to the heatmaps are not prominent enough to read; they are blurry. Please fix that. 

3. Add a comment on whether multi KRAS (on) inhibitor RMC-6236 might play any role in the instances mentioned in this paper. Add a line in the discussion on this topic.

4. Add scale bars in Fig 3 B pictures. 

Reviewer 2 Report

Comments and Suggestions for Authors

This article focused on a cohort study of 27 patients with KRAS WT PDAC and analyzed the molecular characteristics based on transcriptomic profile. The authors also reported effects of therapeutic treatment targeting TFG-MET rearrangement. The patients studied in this research is limited to 27 patients and below are my comments:

1.       Some results in 3.1 are not demonstrated in Table 1: line 138 (“Eight (42%) had Stage IV disease at diagnosis with 15 (79%) additional 138 patients”); line 143(“5 received gemcitabine and nab-paclitaxel with mOS of 15”)

2.       In 3.3, RNA seq analysis has identified two clusters characterized by different KRAS signaling. What are the conclusions based on these finding? What could cause the differences in KRAS activity in the KRAS WT cohort?

3.       In section 3.4, the patient-derived organoid is generated from the biopsy sample from a male patient. What are the standards of selecting the patient? What is the KRAS signaling profile of the patient based on the findings in figure 2 and how is it related to the TFG-MET translocation? The authors should provide more information regarding the importance of studying MET translocation, including the clinical relevance and its roles associated with KRAS.

4.       Line 209 (“Given this mutation may be associated with increased MET signaling”) mentioned that MET translocation is associated with increased MET signaling. Experiment should be performed to support this before the sensitivity test. A similar transcriptomic analysis can be provided based on the data in figure 2.

5.       The reagents information in methods section (2.7) should include catalog number.

Round 2

Reviewer 1 Report

Comments and Suggestions for Authors

All concerns addressed and ready for acceptance. 

Reviewer 2 Report

Comments and Suggestions for Authors

No comment for the current manuscript.